# TRANSFORMER-XH: MULTI-EVIDENCE REASONING WITH EXTRA HOP ATTENTION

**Chen Zhao**[*]
University of Maryland, College Park
chenz@cs.umd.edu

**Chenyan Xiong, Corby Rosset, Xia Song, Paul Bennett, and Saurabh Tiwary**
Microsoft AI & Research
cxiong, corosset, xiaso,
pauben, satiwary@microsoft.com

## ABSTRACT

Transformers have achieved new heights modeling natural language as a sequence of text tokens. However, in many real world scenarios, textual data inherently exhibits structures beyond a linear sequence such as trees and graphs; many tasks require reasoning with evidence scattered across multiple pieces of texts. This paper presents Transformer-XH, which uses eXtra Hop attention to enable intrinsic modeling of structured texts in a fully data-driven way. Its new attention mechanism naturally "hops" across the connected text sequences in addition to attending over tokens within each sequence. Thus, Transformer-XH better conducts joint multi-evidence reasoning by propagating information between documents and constructing global contextualized representations. On multi-hop question answering, Transformer-XH leads to a simpler multi-hop QA system which outperforms previous state-of-the-art on the HotpotQA FullWiki setting. On FEVER fact verification, applying Transformer-XH provides state-of-the-art accuracy and excels on claims whose verification requires multiple evidence.

## 1 INTRODUCTION

Transformers effectively model natural language in *sequential* form (Vaswani et al., 2017; Dai et al., 2019; Devlin et al., 2019; Yang et al., 2019). Nevertheless, in many NLP tasks, text does not simply appear as a linear sequence of tokens but rather carries meaningful structure in the form of paragraphs, headings, and hyperlinks. These structures can be represented abstractly as trees or graphs with nodes and edges; and the tasks can be performed as joint reasoning on these more general structures as input. Multi-hop question answering (Yang et al., 2018) is one such task in which structure plays an important role, since the evidence required to formulate the answer is scattered across multiple documents, requiring systems to jointly reason across links between them.

Recent approaches leverage pre-trained Transformers (e.g., BERT) for multi-hop question answering (QA) by converting the structural reasoning task into sub-tasks that model flat sequences. For example, Min et al. (2019b) decompose a multi-hop question into a series of single-hop questions; Ding et al. (2019) conduct several steps of single-hop reading comprehension to simulate the multi-hop reasoning. The hope is that additional processing to fuse the outputs of the sub-models can recover all the necessary information from the original structure. While pre-trained Transformer language models have shown improvements on multi-hop QA, manipulating the inherent structure of the problem to fit the rigid requirements of out-of-the-box models can introduce problematic assumptions or information loss.

This paper presents Transformer-XH (meaning eXtra Hop), which upgrades Transformers with the ability to natively represent structured texts. Transformer-XH introduces extra hop attention in its layers that connects different text pieces following their inherent structure while also maintaining the powerful pre-trained Transformer abilities over each textual piece individually. Our extra hop attention enables 1) a more global representation of the evidence contributed by each piece of text as it relates to the other evidence, and 2) a more natural way to jointly reason over an evidence graph by propagating information along edges necessary to complete the task at hand.

---

[*]Work done while interning at Microsoft.

We apply Transformer-XH to two multi-evidence reasoning tasks: Hotpot QA, the multi-hop question answering task (Yang et al., 2018), and FEVER, the fact verification benchmark whose claims often require multiple pieces of evidence to support (Thorne et al., 2018). Rather than decomposing the task into a series of sub-tasks to fit the constraints of pre-trained Transformers, Transformer-XH is a solution that fits the problem as it naturally occurs. It is a single model that represents and combines evidence from multiple documents to conduct the reasoning process. On HotpotQA's FullWiki setting, which requires strong multi-hop reasoning capability (Min et al., 2019b; Jiang & Bansal, 2019), Transformer-XH outperforms CogQA (Ding et al., 2019), the previous start-of-the-art, by 12 points on answer F1. On FEVER 1.0 shared task, Transformer-XH outperforms GEAR, the Graph Neural Network based approach significantly. On both applications, Transformer-XH beats the contemporary BERT based pipeline SR-MRS (Nie et al., 2019), by 2-3 points.

The results follow from our simple yet effective design, with one unified model operating over the inherent structure of the task, rather than melding the outputs from disparate sub-tasks adapted to the sequential constraints of pre-trained Transformers. Our ablation studies demonstrate Transformer-XH's efficacy on questions that are known to require multi-hop reasoning (Min et al., 2019b) and on verifying multi-evidence claims (Liu et al., 2019b). Our analyses confirm that the source of Transformer-XH's effectiveness success is due to the eXtra Hop attention's ability to fuse and propagate information across multiple documents.[1]

## 2 Model

This section first discusses preliminaries on sequential Transformers, then we show how we incorporate eXtra hop attention to create Transformer-XH.

### 2.1 Preliminaries

Transformers represent a sequence of input text tokens $X = \{x_1, ..., x_i, ..., x_n\}$ as contextualized distributed representations $H = \{h_1, ..., h_i, ..., h_n\}$ (Vaswani et al., 2017). This process involves multiple stacked self-attention layers that converts the input $X$ into $\{H^0, H^1, ..., H^l, ...H^L\}$, starting from $H^0$, the embeddings, to the final layer of depth $L$.

The key idea of Transformer is its attention mechanism, which calculates the $l$-th layer output $H^l$ using the input $H^{l-1}$ from the previous layer:

$$H^l = \text{softmax}(\frac{Q \cdot K^T}{\sqrt{d_k}}) \cdot V^T, \tag{1}$$

$$Q^T; K^T; V^T = W^q \cdot H^{l-1}; W^k \cdot H^{l-1}; W^v \cdot H^{l-1}. \tag{2}$$

It includes three projections on the input $H^{l-1}$: Query (Q), Key (K), and Value (V).

Specifically, the slices of token $h_i^l$ in Eqn.(2) is:

$$h_i^l = \sum_j \text{softmax}_j(\frac{q_i^T \cdot k_j}{\sqrt{d_k}}) \cdot v_j, \tag{3}$$

which first calculates its attention to all other tokens $j$ in the sequence and then combines the token values $v_j$ into a new representation $h_i^l$, using the normalized attention weights. Multiple attentions can be used in one Transformer layer and concatenated as multi-head attention (Vaswani et al., 2017). The architecture is stacked to form rather deep networks, which leads to significant success of large pre-trained Transformer models (Devlin et al., 2019; Liu et al., 2019a).

A challenge of Transformer is that its attention is calculated over all token pairs (Eqn. 3), which is hard to scale to long text sequences. Transformer-XL (eXtra Long) addresses this challenge by breaking down longer texts, e.g., a multi-paragraph document, into a sequence of text segments: $\{X_1, ..., X_\tau, ..., X_\zeta\}$, and propagates the information between adjacent text segments using the following attention:

$$\tilde{H}_\tau^{l-1} = [\text{Freeze}(H_{\tau-1}^{l-1}) \circ H_\tau^{l-1}]. \tag{4}$$

---

[1] Code available at `https://aka.ms/transformer-xh`.

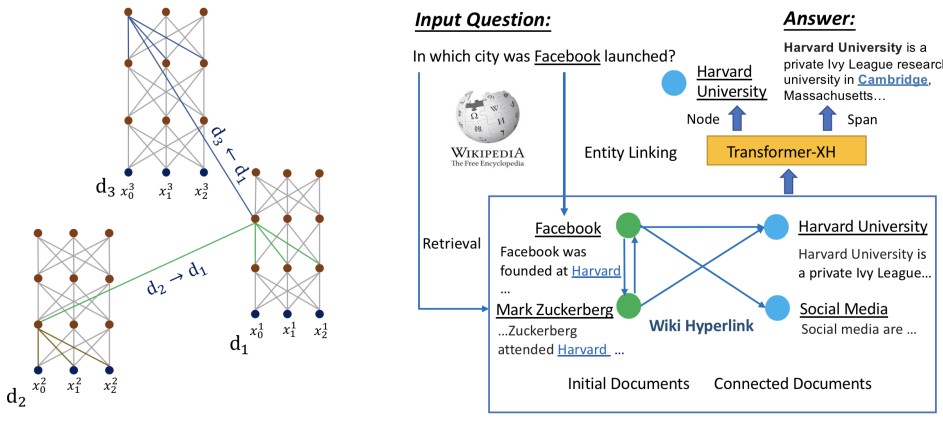

(a) Hop attention on the path $d_2 \to d_1 \to d_3$.     (b) Transformer-XH in Multi-hop QA

Figure 1: The eXtra Hop attention in Transformer-XH (a) and its application to multi-hop QA (b).

It concatenates ($\circ$) the representation of the previous segment $H_{\tau-1}^{l-1}$ to the current segment as segment level recurrences. The new representation $\tilde{H}_{\tau}^{l-1}$ includes the information from previous segment and is integrated in the new attention mechanism:

$$\tilde{Q}^T; \tilde{K}^T; \tilde{V}^T = W^q \cdot H_{\tau}^{l-1}; W^k \cdot \tilde{H}_{\tau}^{l-1}; W^v \cdot \tilde{H}_{\tau}^{l-1}. \tag{5}$$

The attention over the previous segment allows Transformer-XL to effectively model long form text data recurrently as a sequence of text chunks (Dai et al., 2019).

Nevertheless, in many scenarios, the text segments are organized in nontrivial structures beyond a linear sequence. For example, documents are connected by hyperlinks in a graphical structure that does not readily simplify to form a linear sequence, prohibiting Transformer-XL's recurrent approach.

## 2.2 TRANSFORMER-XH WITH EXTRA HOP ATTENTION

Transformer-XH models structured text sequence by linking them with eXtra Hop attention following their original structure. As illustrated in Figure 1a, to model three connected documents $d_2 \to d_1 \to d_3$, Transformer-XH uses eXtra Hop attention to propagate information along the graph edges, enabling information sharing between connected text sequence.

Formally, the structured text data includes a set of nodes, $\mathcal{X} = \{X_1, ..., X_\tau, ...X_\zeta\}$, each corresponding to a text sequence, and an edge matrix $E$, which includes the connections (e.g., links) between them. The goal is to learn representations $\mathcal{H} = \{\tilde{H}_1, ..., \tilde{H}_\tau, ...\tilde{H}_\zeta\}$, that incorporate not only the local information in each sequence $X$, but also the global contexts on the entire structured text $\{\mathcal{X}, E\}$.

Transformer-XH achieves this by two attention mechanisms: in-sequence attention and eXtra Hop attention. The *in-sequence attention* is the same as vanilla Transformer: in layer $l$, token $i$ gathers information from other tokens inside the same text piece $\tau$:

$$h_{\tau,i}^l = \sum_j \text{softmax}_j\left(\frac{q_{\tau,i}^T \cdot k_{\tau,j}}{\sqrt{d_k}}\right) \cdot v_{\tau,j}. \tag{6}$$

The *eXtra Hop attention* uses the first token in each sequence – the added special token "[CLS]" – as an "attention hub", which attends on all other connected nodes' hub token. In layer $l$, the $\tau$-th text sequence attends over another text sequence $\eta$ if there is an edge between them ($e_{\tau\eta} = 1$):

$$\hat{h}_{\tau,0}^l = \sum_{\eta;e_{\tau\eta}=1} \text{softmax}_\eta\left(\frac{\hat{q}_{\tau,0}^T \cdot \hat{k}_{\eta,0}}{\sqrt{d_k}}\right) \cdot \hat{v}_{\eta,0}. \tag{7}$$

Node $\tau$ calculates the attention weight on its neighbor $\eta$ using hop query $\hat{q}_{\tau,0}$ and key $\hat{k}_{\eta,0}$. Then it uses the weights to combine its neighbors' value $\hat{v}_{\eta,0}$ and forms a globalized representation $\hat{h}_{\tau,0}^l$.

The two attention mechanism are combined to form the new representation of layer $l$:

$$\tilde{h}_{\tau,0}^l = \text{Linear}([h_{\tau,0}^l \circ \hat{h}_{\tau,0}^l]), \tag{8}$$

$$\tilde{h}_{\tau,i}^l = h_{\tau,i}^l; \forall i \neq 0. \tag{9}$$

Note that the non-hub tokens ($i \neq 0$) still have access to the hop attention in the previous layer through Eqn. (6).

One layer of eXtra Hop attention can be viewed as single-step of information propagation along edges $E$. For example, in Figure 1a, the document node $d_3$ updates its representation by gathering information from its neighbor $d_1$ using the hop attention $d_1 \rightarrow d_3$. When multiple Transformer-XH layers are stacked, this information in $d_1$ includes both $d_1$'s local contexts from its in-sequence attention, and cross-sequence information from the hop attention $d_2 \rightarrow d_1$ of the $l-1$ layer. Hence, an L-layer Transformer-XH can attend over information from up to L hops away.

Together, three main properties equip Transformer-XH to effectively model raw structured text data: the propagation of information (values) along edges, the importance of that information (hop attention weights), and the balance of in-sequence and cross-sequence information (attention combination). The representations learned in $\mathcal{H}$ can innately express nuances in structured text that are required for complex reasoning tasks such as multi-hop QA and natural language inference.

## 3 Application to Multi-Hop Question Answering

This section describes how Transformer-XH applies to multi-hop QA. Given a question $q$, the task is to find an answer span $a$ in a large open-domain document corpus, e.g. the first paragraph of all Wikipedia pages. By design, the questions are complex and often require information from multiple documents to answer. For example, in the case shown in Figure 1b, the correct answer "Cambridge" requires combining the information from both the Wikipedia pages "Facebook" and "Harvard University". To apply Transformer-XH in the open domain multi-hop QA task, we first construct an evidence graph and then apply Transformer-XH on the graph to find the answer.

**Evidence Graph Construction.** The first step is to find the relevant candidate documents $D$ for the question $q$ and connect them with edges $E$ to form the graph $G$. Our set $D$ consists of three sources. The first two sources are from canonical information retrieval and entity linking techniques:

$D_{ir}$: the top 100 documents retrieved by DrQA's TF-IDF on the question (Chen et al., 2017).

$D_{el}$: the Wikipedia documents associated with the entities that appear in the question, annotated by entity linking systems: TagMe (Ferragina & Scaiella, 2010) and CMNS (Hasibi et al., 2017).

For better retrieval quality, we use a BERT ranker (Nogueira & Cho, 2019) on the set $D_{ir} \cup D_{el}$ and keep the top two ranked ones in $D_{ir}$ and top one per question entity in $D_{el}$. Then the third source $D_{exp}$ includes all documents connected to or from any top ranked documents via Wikipedia hyperlinks (e.g., "Facebook" $\rightarrow$ "Harvard University").

The final graph comprises all documents from the three sources as nodes $\mathcal{X}$. The edge matrix $E$ is flexible. We experiment with various edge matrix settings, including directed edges along Wikipedia links, i.e. $e_{ij} = 1$ if there is a hyperlink from document $i$ to $j$, bidirectional edges along Wiki links, and fully-connected graphs, which rely on Transformer-XH to learns the edge importance.

Similar to previous work (Ding et al., 2019), the textual representation for each node in the graph is the [SEP]-delimited concatenation of the question, anchor text (the text in the hyperlink in parent nodes pointing to the child node), and the paragraph itself. More details on the evidence graph construction are in Appendix A.1.

**Transformer-XH on Evidence Graph.** Transformer-XH takes the input nodes $\mathcal{X}$ and edges $E$, and produces the global representation of all text sequences:

$$\mathcal{H}^L = \text{Transformer-XH}(\mathcal{X}, E). \tag{10}$$

Then we add two task-specific layers upon the last layer's representation $\mathcal{H}^L$: one auxiliary layer to predict the relevance score of the evidence node, and one layer to extract the answer span within it:

$$p(\text{relevance}|\tau) = \text{softmax}(\text{Linear}(\tilde{h}^L_{\tau,0})); \tag{11}$$

$$p(\text{start}|\tau,i), p(\text{end}|\tau,j) = \text{softmax}(\text{Linear}(\tilde{h}^L_{\tau,i})), \text{softmax}(\text{Linear}(\tilde{h}^L_{\tau,j})). \tag{12}$$

The final model is trained end-to-end with cross-entropy loss for both tasks in a multi-task setting. During inference, we first select the document with the highest relevance score, and then the start and end positions of the answer within that document.

## 4 Application to Fact Verification

This section describes how Transformer-XH applies to the fact verification task in FEVER Thorne et al. (2018). Given a claim and a trustworthy background corpus, i.e. Wikipedia, the task is to verify whether the evidence in the corpus SUPPORTS, REFUTES, or there is NOT ENOUGH INFO to verify the claim. Similar to multi-hop QA, the first step is to construct an evidence graph using the text pieces in the background corpus and then Transformer-XH can be easily applied to conduct reasoning on these evidence pieces.

**Evidence Graph Construction.** Many previous FEVER systems first retrieve the evidence sentences for the claim and then reason verify it (Nie et al., 2019; Zhou et al., 2019). This first step is similar as the retrieval stage in Hotpot QA. And the second step is a multi-evidence reasoning task, where Transformer-XH is applied.

We keep the evidence sentence retrieval step consistent with previous methods. The sentence retrieval results of SR-MRS is not yet released at the time of our experiments, thus we instead use the BERT-based retrieval results from another contemporary work (Liu et al., 2019b).

We construct the evidence graph using the top five sentences from Liu et al. (2019b) as the nodes $\mathcal{X}$ and fully connected edges $E$. Following Liu et al. (2019b), the representation of each node is the concatenation of the claim, the Wikipedia title (entity name) of the document that includes the sentence, and the evidence sentence.

**Transformer-XH on Evidence Graph.** Transformer-XH takes the evidence graph $\{X, E\}$ and learns to verify the claim to three categories: $y \in \{\text{SUPPORT, REFUSE, NOT ENOUGH EVI-DENCE}\}$. Similar to the application in Hotpot QA, it first produces the global representation of the graph:

$$\mathcal{H}^L = \text{Transformer-XH}(\mathcal{X}, E). \tag{13}$$

Then two task-specific layers are added upon the last layer. The first layer conducts the fact prediction per node using the "[CLS]" token:

$$p(y|\tau) = \text{softmax}(\text{Linear}(\tilde{h}^L_{\tau,0})). \tag{14}$$

The second layer learns to measure the importance of each node in the graph:

$$p(s|\tau) = \text{softmax}(\text{Linear}(\tilde{h}^L_{\tau,0})), \tag{15}$$

The node level predictions and node importance are combined to the final prediction for the claim:

$$p(y|\mathcal{X}, E) = \sum_\tau p(s|\tau) \cdot p(y|\tau). \tag{16}$$

Similar to the Hotpot QA scenario, we use multi-task learning that combines the node prediction task and the claim verification task. The first task uses the evidence sentence label provided by FEVER and cross-entropy loss on Eqn. 15. The second task uses the final verification label with cross-entropy loss on Eqn. 16.

## 5 Experimental Methodologies

Our experiments are conducted on Hotpot QA, the multi-hop question answering benchmark Yang et al. (2018), and FEVER, the fact verfication benchmark Thorne et al. (2018).

## 5.1 MULTI-HOP QUESTION ANSWERING ON HOTPOT QA

**Dataset.** HotpotQA includes 112k crowd-sourced questions designed to require multiple pieces of textual evidence, which are the first paragraphs of Wikipedia pages. It has two type of questions: bridge question require hopping via an outside entity, and comparison question compare a property of two entities. There are two settings in HotpotQA. The Distractor setting provides golden evidence paragraphs together with TF-IDF retrieved negatives. The FullWiki setting requires systems to retrieve evidence paragraphs from the full set of Wikipedia articles.

We focus on FullWiki setting since previous research found that the negative documents in Distractor may be too weak and mitigate the need for multi-hop reasoning (Min et al., 2019b). There are 90k Train, 7k Dev and 7k Test questions. The ground truth answer and supporting evidence sentences in Train and Dev sets are provided. Test labels are hidden; only one submission is allowed to the leaderboard per 30 days[2]. We evaluate our final model on Test and conduct ablations on Dev.

**Metrics.** We use official evaluation metrics of HotpotQA: exact match (EM) and F1 on answer (Ans), supporting facts (Supp), and the combination (Joint). The supporting facts prediction is an auxiliary task that evaluates model's ability to find the evidence sentences. Joint EM is the product of the two EM result. Joint F1 first multiplies the precision and recall from Ans and Supp, then combines the Joint precision and recall to F1.

**Baseline.** The main baselines include Cognitive QA (CogQA, Ding et al. (2019)) and Semantic Retrieval MRS (SR-MRS, Nie et al. (2019)). CogQA uses several fine-tuned BERT machine reading comprehension (MRC) models to find hop entities and candidate spans, and then uses a BERT based Graph Convolution Network to rank the candidate spans. SR-MRS is a contemporary work and was the previous leaderboard rank one. It is a BERT based pipeline and uses fine-tuned BERT models to first rank the documents (twice), then to rank sentences to find supporting facts, and finally conducts BERT MRC on the concatenated evidence sentences.

We also re-implement CogQA and upgrade its IR with our BERT IR model (BERT on $D_{ir} \cup D_{el}$, same as Transformer-XH), for fair comparisons. We include other approaches on the FullWiki setting: Official Baseline (Yang et al., 2018), MUPPET (Feldman & El-Yaniv, 2019), QFE (Nishida et al., 2019), and DecompRC (Min et al., 2019a),

**Implementation Details.** The in-sequence attention and other standard Transformer components in Transformer-XH are initialized by the pre-trained BERT base model (Devlin et al., 2019). The extra hop attention parameters are initialized randomly and trained from scratch. The final model uses three hop steps. For bridge questions, we build the evidence graph described in Section 3. And for comparison questions, we build the fully-connected graph on the set $D_{ir} \cup D_{el}$ and train Transformer-XH separately. We leave more implementation details in the Appendix.

## 5.2 FACT VERIFICATION ON FEVER

**Dataset.** The FEVER task provides a claim sentence and requires the system to classify it into three categories: SUPPORTS, REFUTES, and NOT ENOUGH INFO, using the Wikipedia corpus as the evidence source. It provides 185,455 claims with manual labels and uses the Wikipedia dump in June 2017 which includes 5.4 million documents.

**Metrics.** There are two official evaluation metrics in FEVER: Label Accuracy (LA), which evaluates the classification accuracy of the verification labels, and FEVER Score, which evaluates both the correctness of the evidence sentences used in verification and the LA. The latter is close to Joint EM in Hotpot QA and is the main metric. We use the official evaluation scripts from FEVER task and we refer to Thorne et al. (2018) for more details of this task.

**Experimental Setups.** We follow the experiment settings used by previous research in FEVER 1.0 shared task, i.e. Nie et al. (2019), Zhou et al. (2019), and Liu et al. (2019b). Similar as Liu et al. (2019b), we also split the data into single and multi evidence categories and evaluate Transformer-XH on the two splits.

---

[2]https://hotpotqa.github.io/

| | Dev | | | | | | Test | | | | | |
|---|---|---|---|---|---|---|---|---|---|---|---|---|
| | Ans | | Supp | | Joint | | Ans | | Supp | | Joint | |
| | EM | F1 | EM | F1 | EM | F1 | EM | F1 | EM | F1 | EM | F1 |
| Official Baseline (Yang et al., 2018) | 23.9 | 32.9 | 5.1 | 40.9 | 47.2 | 40.8 | 24.0 | 32.9 | 3.9 | 37.7 | 1.9 | 16.2 |
| DecompRC (Min et al., 2019a) | - | 43.3 | - | - | - | - | 30.0 | 40.7 | - | - | - | - |
| QFE (Nishida et al., 2019) | - | - | - | - | - | - | 28.7 | 38.1 | 14.2 | 44.4 | 8.7 | 23.1 |
| MUPPET (Feldman & El-Yaniv, 2019) | 31.1 | 40.4 | 17.0 | 47.7 | 11.8 | 27.6 | 30.6 | 40.3 | 16.7 | 47.3 | 10.9 | 27.0 |
| CogQA (Ding et al., 2019) | 37.6 | 49.4 | 23.1 | 58.5 | 12.2 | 35.3 | 37.1 | 48.9 | 22.8 | 57.7 | 12.4 | 34.9 |
| SR-MRS* (Nie et al., 2019) | 46.5 | 58.8 | 39.9 | 71.5 | 26.6 | 49.2 | 45.3 | 57.3 | 38.7 | 70.8 | 25.1 | 47.6 |
| CogQA (w. BERT IR) [Ours] | 44.8 | 57.7 | 29.2 | 62.8 | 18.5 | 43.4 | - | - | - | - | - | - |
| Transformer-XH | **54.0** | **66.2** | **41.7** | **72.1** | **27.7** | **52.9** | **51.6** | **64.1** | **40.9** | **71.4** | **26.1** | **51.3** |

Table 1: Results (%) on HotpotQA FullWiki Setting. Dev results of previous methods are reported in their papers. Test results are from the leaderboard. Contemporary method is marked by *.

| | Question Type | | | | Reasoning Type | | | |
|---|---|---|---|---|---|---|---|---|
| | Comparison (1487) | | Bridge (5918) | | Single-Hop (3426) | | Multi-Hop (3979) | |
| | EM | F1 | EM | F1 | EM | F1 | EM | F1 |
| CogQA | 43.3 | 51.1 | 36.1 | 49.0 | 45.1 | 61.1 | 31.1 | 39.4 |
| SR-MRS* | 62.0 | 68.9 | 42.4 | 56.1 | 52.3 | 68.4 | 41.3 | 50.3 |
| CogQA (w. BERT IR) | 54.1 | 60.9 | 42.4 | 56.9 | 52.0 | 69.3 | 38.6 | 47.8 |
| Transformer-XH (w. BERT IR) | 59.9 | 65.8 | **52.4** | **66.3** | **62.2** | **78.3** | **46.8** | **55.7** |
| Transformer-XH (w. SR-MRS) | **64.3** | **70.7** | 47.9 | 62.3 | 58.1 | 74.3 | 45.3 | 55.2 |

Table 2: Dev Ans (%) on different scenarios. Reasoning Types are estimated by Min et al. (2019b) via whether single-hop BERT has non-zero Ans F1. The numbers of questions are shown in brackets.

**Baselines.** The baselines include GEAR (Zhou et al., 2019) and two contemporary work, SR-MRS (Nie et al., 2019) and KGAT (Liu et al., 2019b). SR-MRS uses similar adaptations as Transformer-XH from Hotpot QA to FEVER. GEAR is a graph attention network based approach specially designed for fact verification. KGAT further improves GEAR's GAT by adding the kernel information, and is the previous STOA with BERT base. We also include the BERT Concat baseline Liu et al. (2019b) which concatenates the evidence sentences to a text sequence and applies BERT on it.

**Implementation Details.** We use the retrieval result from Liu et al. (2019b) and connect all sentences as a fully connected graph. We follow similar parameter settings as Hothot QA. We use pre-trained BERT base model to initialize the Transformer components. The extra hop attention parameters are initialized randomly and trained from scratch, and three hop steps are used. We train Transformer-XH for two epochs.

## 6 EVALUATION RESULTS

This section first presents the evaluation results on HotpotQA and FEVER. Then it conducts ablation studies, analyses, and case studies on HotpotQA to understand the effectiveness of Transformer-XH.

### 6.1 OVERALL RESULT

**HotpotQA** FullWiki results are presented in Table 1. Transformer-XH outperforms all previous methods by significant margins. Besides strong results, Transformer-XH's ability to natively represent structured data leads to much simpler QA system. Previously, in order to utilize pre-trained BERT, Hotpot QA approaches adapted the multi-hop reasoning task to comprise multiple sub-tasks. For example, given the retrieved documents, CogQA (w. BERT IR) first leverages one BERT MRC model to find hop entities and then another BERT MRC to find candidate answer spans. After that, it ranks the candidate spans using a BERT based GAT, which is the only structure modeling step. In comparison, Transformer-XH is a unified model which directly represents structured texts and integrates BERT weights.

| | Dev | | Test | | Single Evidence | | Multi Evidence | |
|---|---|---|---|---|---|---|---|---|
| | LA | FEVER | LA | FEVER | LA | FEVER | LA | FEVER |
| BERT Concat (Liu et al., 2019b) | 73.67 | 68.89 | 71.01 | 65.64 | - | - | - | - |
| GEAR/GAT (Zhou et al., 2019) | 74.84 | 70.69 | 71.60 | 67.10 | 79.79 | 77.42 | 66.12 | 38.21 |
| SR-MRS* (Nie et al., 2019) | 75.12 | 70.18 | 72.56 | 67.26 | - | - | - | - |
| KGAT* (Liu et al., 2019b) | 78.02 | **75.88** | **72.81** | **69.40** | 80.33 | 78.07 | 65.92 | 39.23 |
| Transformer-XH | **78.05** | 74.98 | 72.39 | 69.07 | **81.84** | **81.31** | **86.58** | **58.47** |

Table 3: FEVER Results. Contemporary work is marked by *. Single and Multi Evidence are results on Dev claims on which one or multiple sentences are labeled as evidence.

Table 2 further inspects model performances on the Dev set by question types and reasoning types. Transformer-XH significantly outperforms all baselines on bridge questions which require more multi-hop reasoning. And on the "multi-hop" questions, Transformer-XH has higher relative gains (39% over CogQA on EM) than the "single-hop" questions (27%), demonstrating its stronger multi-hop reasoning capability. We further study this in Section 6.3.

To further investigate the reasoning ability of Transformer-XH, we replace our retrieval pipeline with the top retrieved documents from the SR-MRS pipeline. More specifically, we use the top retrieved documents from SR-MRS to construct Transformer-XH's evidence graph while keeping all else constant. The resulting system, Transformer-XH (w. SR-MRS), outperforms SR-MRS's multi-step BERT based reasoning on all metrics and question types. Transformer-XH's effectiveness is robust with multiple IR systems.

**FEVER** fact verification results are shown in Table 3. Transformer-XH outperforms SR-MRS by 4 FEVER score on Dev and 1.8 on Test. It performs on par with KGAT. More importantly, Transformer-XH excels at verifying claims that require multiple pieces of evidence–outperforming the contemporary work KGAT by 20 FEVER scores on the multi-evidence claims, a 49% relative improvement. Compared to KGAT, Transformer-XH mainly loses on the "not enough evidence" category which is neither single nor multi evidence. This is an artifact the FEVER task which our system is not specifically designed for.

This result also demonstrates Transformer-XH's generality on tasks with multiple text inputs not in sequential formats. The only difference between Transformer-XH when applied on multi-hop QA and FEVER is the last (linear) task specific layer; it provides similar or better performances over contemporary approaches that were specifically designed for the fact verification task. Due to space constraints and the consistent effectiveness of Transformer-XH on the two applications, the rest experiments mainly used HotpotQA to analyze the behavior of Transformer-XH.

## 6.2 ABLATION STUDIES

**Model Variations.** We show the results of different model variations on the top left of Table 4. *Single-Hop BERT* uses BERT MRC model on each document individually, which significantly decreases the accuracy, confirming the importance of multi-hop reasoning in FullWiki setting (Min et al., 2019a). *GAT + BERT* first uses Graph Attention Network (Veličković et al., 2018) on the evidence graph to predict the best node; then it uses BERT MRC on the best document. It is 10% worse than Transformer-XH since the MRC model has no access to the information from other documents. *No Node Prediction* eliminates the node prediction task and only trains on span prediction task; the accuracy difference shows node prediction task helps the model training.

**Graph Structures.** We show Transformer-XH's performance with different graph structures on the bottom left of Table 4. *Bidirectional Edges* adds reverse edges along the hyperlinks; *Fully Connected Graph* connects all document pairs; *Node Sequence* randomly permutes the documents and connects them into a sequence to simulate the Transformer-XL setting. Both *Bidirectional Links* and *Fully Connected Graph* have comparable performance with the original graph structure. Transformer-XH is able to learn meaningful connections using its hop attentions and is less dependent on the pre-existing graph structural. The fully connected graph can be used if there is no strong edge patterns available in the task. However, the performance drops significantly on *Node Sequence*, showing that structured texts cannot be treated as a linear sequence which cuts off many connections.

| Model Ablation | Dev Ans | | Hop Steps | Dev Ans | |
|---|---|---|---|---|---|
| | EM | F1 | | EM | F1 |
| Single-Hop BERT MRC on Individual Documents | 31.3 | 42.2 | One Hop | 50.3 | 64.6 |
| GAT (Node Prediction) + BERT (MRC on Best Node) | 48.9 | 61.9 | Two Hops | 51.6 | 66.4 |
| No Node Prediction Multi-Task | 43.2 | 55.3 | Four Hops | 51.4 | 66.1 |
| Bidirectional Edges on Hyperlinks | 50.6 | 65.0 | Five Hops | 50.6 | 64.7 |
| Fully Connected Graph | 51.0 | 65.5 | Six Hops | 50.1 | 64.2 |
| Node Sequence (Bidirectional Transformer-XL) | 14.1 | 20.7 | Transformer-XH | **52.4** | **66.3** |

Table 4: Ablation studies on the bridge questions on Dev answer accuracy (%), including model components (top left), graph structures (bottom left), and hop steps (right). Transformer-XH's full model uses three hop step and unidirectional Wiki link graph.

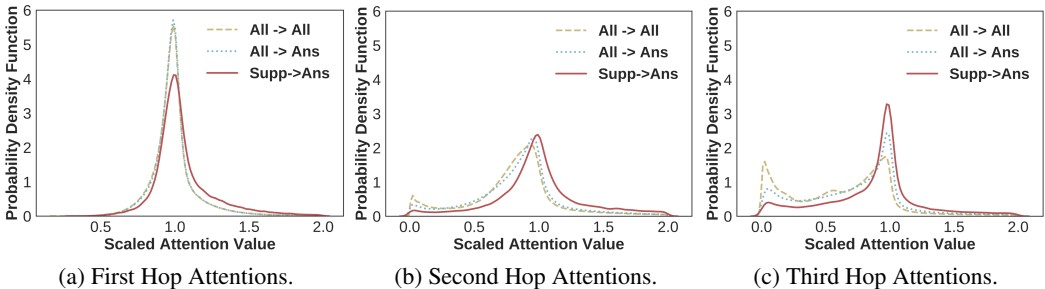

(a) First Hop Attentions.    (b) Second Hop Attentions.    (c) Third Hop Attentions.

Figure 2: Distributions of learned attention weights of three hops on three groups: From All (Node) → (to) All, All → (to) Ans (ground truth answer node), and Supp (nodes with the supporting facts) → (to) Ans. X-axes are attention values scaled by number of nodes.

**Hop Steps.** Recall that a Transformer-XH layer with extra hop attention corresponds to one information propagation (hop) step in the graph. Thus Transformer-XH with last K layers conducts K-step attention hops in the graph. We show results with different K on the right side of Table 4. Transformer-XH reaches its peak performance with three hops (our full-model). This is expected as most Hotpot QA questions can be answered by two documents (Yang et al., 2018).

## 6.3 HOP ATTENTION ANALYSIS

This experiment analyzes the hop attentions using our full-model (three-hop) on the fully connected graph to study their behavior without pre-defined structure. Figure 2 plots the distributions of the learned hop attentions on the Dev set. It shows a strong shift away from the normal distribution with more hops. Transformer-XH learns to distinguish different nodes after multi-hop attention: the attention score becomes a bimodal distribution after three hops, ignoring some non-useful nodes. Transformer-XH also learns to focus on meaningful edges: the score is higher on the path Supp→Ans than All→Ans. And the margin is larger as the hop step increases from one to three.

## 6.4 CASE STUDY

Table 5 lists two examples from Transformer-XH and CogQA (w. BERT IR). The first case has a clear evidence chain "2011/S/S"→"Winner"→"YG Entertainment"; both methods find the correct answer. However, the second case has too many distractors in the first document. Without additional clues from document 2, it is likely that the single-hop hop entity extraction component in CogQA (w. BERT IR) misses the correct answer document in its candidate sets; and the later structural reasoning component can not recover from this cascade error. In comparison, Transformer-XH finds the correct answer by combining the evidence with the hop attentions between the two evidence pieces. We leave more positive and negative cases in Appendix A.5.

| Q: 2014 S/S is the debut album of a South Korean boy group that was formed by who?
**Document 1**: 2014 S/S is the debut album of South Korean group Winner.
**Document 2**: Winner is a South Korean boy group formed in 2013 by **YG Entertainment**.
**Transformer-XH**: YG Entertainment ✓
**CogQA (w. BERT IR)**: YG Entertainment ✓ | Q: Which man who presented 2022 FIFA World Cup bid was born on October 22, 1930?
**Document 1**: 2022 FIFA World Cup bid was presented by Frank Lowy, Ben Buckley, Quentin Bryce and Elle Macpherson.
**Document 2**: **Frank Lowy** (born 22 October 1930), is an Australian-Israeli businessman and Chairman of Westfield Corporation.
**Transformer-XH**: Frank Lowy ✓
**CogQA (w. BERT IR)**: Quentin Bryce ✗ |
|---|---|

Table 5: Examples of Transformer-XH and BERT pipeline results in Hotpot QA.

## 7 RELATED WORK

HotpotQA's FullWiki task is a combination of open-domain QA (Chen et al., 2017) and multi-hop QA (Yang et al., 2018): the questions are designed to require multiple pieces of evidence and these evidence pieces are documents to retrieve from Wikipedia. It is a challenging combination: The retrieved documents are inevitably noisy and include much stronger distractors than the TF-IDF retrieved documents in the Distractor setting (Min et al., 2019a; Jiang & Bansal, 2019).

Various solutions have been proposed for Hotpot QA (Min et al., 2019b; Feldman & El-Yaniv, 2019; Nishida et al., 2019). These solutions often use complicated pipelines to adapt the multi-hop task into a combination of single-hop tasks, in order to leverage the advantage of pre-trained models. For example, CogQA (Ding et al., 2019) uses two BERT based MRC model to find candidate spans and then another BERT initialized Graph Neural Network (GNN) to rank spans; SR-MRS (Nie et al., 2019) uses three BERT based rankers to find supporting sentences, and then another BERT MRC model on the concatenated sentences to get the answer span. Transformer-XH is a simpler model that directly represents and reasons with multiple pieces of evidence using extra hop attentions.

Fact verification is a natural language inference task while also requires retrieving ("open-domain") and reasoning with multiple text pieces ("multi-evidence") (Thorne et al., 2018; Nie et al., 2019; Liu et al., 2019b). Many recent FEVER systems leverage Graph Neural Networks to combine information from multiple text nodes, while each node text is represented by BERT encodings (Zhou et al., 2019; Liu et al., 2019b). Transformer-XH is a more unified solution that simply includes language modeling as part of its joint reasoning.

In addition to Transformer-XL (Dai et al., 2019), other work is proposed to improve the Transformer architecture on long text sequence. For example, T-DMCA (Liu et al., 2018) splits the sequence into blocks and then the attention merges different blocks. Sparse Transformer (Child et al., 2019) introduces the sparse factorizations of the attention matrix. Transformer-XH shares similar motivation and focuses on multiple pieces of text that are not in sequential forms.

Transformer-XH is also inspired by GNN (Kipf & Welling, 2017; Schlichtkrull et al., 2017; Veličković et al., 2018), which leverages neural networks to model graph structured data for downstream tasks (Sun et al., 2018; Zhao et al., 2020). The key difference is that a "node" in Transformer-XH is a text sequence, and modeling of the structure is conducted jointly with the representation of the text. Transformer-XH combines the Transformer's advantages in understanding text with the power that GNN has in modeling structure.

## 8 CONCLUSION

Transformer-XH and its eXtra Hop attention mechanism is a simple yet powerful adaptation of Transformer to learn better representations of structured text data as it naturally occurs. It innately integrates with pre-trained language models to allow for complex reasoning across multiple textual evidence pieces. When applied to HotpotQA, Transformer-XH significantly shrinks the typical multi-hop QA pipeline, eliminating many cascading errors that arise from the linear sequence input constraints of pre-trained Transformers. The same simplicity also applies to FEVER, with one Transformer-XH all we needed to obtain a much stronger answer accuracy. With its simplicity and efficacy, we envision Transformer-XH will benefit many applications in the near future.

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

# A   APPENDIX

The appendix includes details of the evidence graph construction for Hotpot QA, ablation studies in the BERT IR component, more details and results on Hotpot QA.

## A.1   HOTPOTQA EVIDENCE GRAPH CONSTRUCTION DETAILS

The evidence graph construction includes two stages. The first stage is BERT IR, which extract related documents directly from question. The second stage expands the related documents along Wikipedia links. The first is applied on all questions while the second is only required by Bridge questions.

The first stage uses two methods to find documents. The first method uses DrQA's retrieval system (Chen et al., 2017), which is unsupervised TF-IDF. We keep the top 100 DrQA retrieved ones $D_{ir}$ for each question. The second method uses TagMe (Ferragina & Scaiella, 2010) and CMNS (Hasibi et al., 2017), two commonly used entity linkers, to annotate questions. We keep the TagMe output entity and three highest scored entities per surface form (a phrase in the question linked with entities) from CMNS, and use its corresponding Wikipedia document as $D_{el}$.

We use BERT ranker (Nogueira & Cho, 2019) to re-rank the initial set $D_{ir} \cup D_{el}$. The input to the BERT is the concatenation of question and first paragraph of document:

```
[CLS] Question [SEP] First Paragraph of Document.
```

Then a linear layer is added on the last layer's [CLS] representation to score the relevance of the document. We use BERT base and fine-tune it using the relevance label (from supporting facts) with cross-entropy loss. The top two highest scored documents from $D_{ir}$ and the top one document per entity position (surface form) in $D_{el}$ are kept as the first stage BERT IR documents.

The second stage expands the first stage BERT IR documents by Wikipedia hyperlinks to obtain $D_{exp}$. A document is included if it is linked to or links to a document in the first stage. We use the same BERT ranker to rank $D_{exp}$ and keep the top 15 documents in $D_{exp}$.

The final evidence graph nodes per question includes the top two highest ranked documents in $D_{ir}$, top one per entity name in $D_{el}$, and top 15 from the expanded documents $D_{exp}$.

The comparison questions only require information from two question entities; thus when building the evidence graph we do not expand them (i.e. there is no $D_{exp}$).

The retrieval pipeline is a multi-stage retrieval enhanced with entity linking. It is close to the retrieval system used in SR-MRS (Nie et al., 2019). When using SR-MRS retrieved documents for documents, we use top 10 documents on bridge questions and top two documents on comparison questions.

In the next section, we show that Transformer-XH is robust to different number of documents kept in the evidence graph and performs similarly using the documents retrieved from SR-MRS.

## A.2   ABLATION STUDIES ON DOCUMENT RETRIEVAL

This experiment studies the effectiveness and influence of different retrieval settings. We use different numbers of top K ranked documents from the BERT ranker, run Transformer-XH in the corresponding evidence graph, and evaluate its performance on Bridge questions in the Dev set. We also evaluate the Supporting facts Recall and Answer Recall. Supp Recall evaluates whether the document with the supporting fact is included in the first stage retrieved documents. Ans Recall evaluates whether their exists a document in the evidence graph that includes the ground truth answer. The results are in Table 6.

Our BERT IR system performs better than CogQA's TF-IDF and on par with SR-MRS, as expected. The latter uses a similar retrieval pipeline with our BERT IR system; Transformer-XH is robust on different retrieval settings and keeps its effectiveness when applied on Top 10 documents from SR-MRS (including both stages).

| Method | Supp Recall | Ans Recall | Dev Ans EM | Dev Ans F1 |
|---|---|---|---|---|
| Top 10 TFIDF (CogQA) | 70.8 | n.a. | - | - |
| Top 2 w. BERT IR + Q Entites | 72.3 | 88.1 | **48.7** | **62.6** |
| Top 5 w. BERT IR + Q Entites | 76.5 | 89.6 | 47.1 | 60.8 |
| Top 10 w. BERT IR + Q Entites | **78.9** | **91.1** | 46.6 | 60.3 |
| SR-MRS Top 10 (All Together) | n.a. | 86.1 | 47.9 | 62.3 |

Table 6: Ablation study on the retrieval systems. Top-10 TFIDF is the one used by CogQA (Ding et al., 2019). BERT IR is the retrieval system used by CogQA (w. BERT IR) and Transformer-XH. Top K refers to using the 2/5/10 highest ranked documents from the BERT ranker in the first stage. SR-MRS Top 10 uses the 10 retrieved documents per question provided by Nie et al. (2019). All retrieval methods include entities linked in the question and are expanded along Wiki links, except when evaluating the 1st stage Supp Recall.

## A.3    OTHER HOTPOT QA COMPONENTS

This section describes the other components for HotpotQA dataset. The whole QA system starts with question type classification. We train transformer-XH separately on each type of questions over their evidence graph. Besides answer prediction, we also adopt BERT based model for predicting supporting sentences.

### A.3.1    QUESTION CLASSIFICATION

The first component of our system is to classify the question to bridge and comparison types. We adopt BERT classification fine-tuning setting on HotpotQA questions using the question type labels provided in HotpotQA. The classifier achieves 99.1% accuracy on the dev set. We use the classifier to split the questions into Comparison and Bridge.

### A.3.2    SUPPORTING FACTS CLASSIFICATION

The supporting facts prediction task is to extract all sentences that help get the answer. For bridge question, these sentences usually cover different pieces of questions. And for comparison questions, the supporting facts are the properties of two question entities. We design one model architecture for this task, but we train two models on each type to reflect the inherent difference.

We use BERT as our base model and on top of BERT, we conduct multi-task learning scheme. The first task is document relevance prediction, similar as Transformer-XH, we add a linear layer on the [CLS] token of BERT to predict the relevance score. The other task is sentence binary classification, we concatenate the first and last token representation of each sentence in the document through a linear layer, the binary output decides whether this sentence is supporting sentence.

**Bridge question supporting facts prediction**    For bridge questions, we predict supporting facts after answer prediction from Transformer-XH to resume the inference chain. We start by predicting supporting facts in the answer document. The other document is chosen from the parents of the answer document in the evidence graph. [3]. Compare with the contemporary model Nie et al. (2019), which does not limit the search space along the inference chain (i.e., the answer document may not be relevant to the other supporting page), our method more naturally fits the task purpose.

**Comparison question supporting facts prediction**    For comparison questions, after extracting the first step documents $D$, we simply run this supporting facts prediction model to select the top-2 documents, and predict the corresponding supporting facts.

### A.3.3    TRAINING DETAILS

We use DGL (Wang et al., 2019) for implementing Transformer-XH and CogQA (w. BERT IR) with batch size 1 (i.e., one graph for each batch), and keep the other parameters same as default BERT setting. We train Transformer-XH separately on two different types of questions, following previous

---

[3]If the answer document does not have parent node, we choose from all documents

| id | Example | Explanation |
|---|---|---|
| 1(+) | **Q**: In which year was the King who made the 1925 Birthday Honours born? 
 **P**: 1865 ✓ 
 **Document 1**: The 1925 Birthday Honours were appointments by King George V to various orders and honours. 
 **Document 2**: George V (3 June 1865 – 20 January 1936) was King of the United Kingdom. | With necessary evidence available, Transformer-XH conducts multi-hop reasoning, and extracts the correct span. |
| 2(-) | **Q**: Where was the world cup hosted that Algeria qualified for the first time into the round of 16? 
 **A**: Brazil    **P**: Spain ✗ 
 **Document 1 (Algeria at the FIFA World Cup)**: In 2014, Algeria qualified for the first time into the round of 16. 
 **Document 2 (2014 FIFA CUP)**: It took place in Brazil from 12 June to 13 July 2014, after the country was awarded the hosting rights in 2007. | Transformer-XH does not predict the correct answer, since document 1 does not link to any other documents.Thus, the information does not propagate to the correct answer document 2014 FIFA CUP. |
| 3(-) | **Q**: What government position was held by the woman who portrayed Corliss Archer in the film Kiss and Tell? 
 **A**: Chief of Protocol    **P**: ambassador ✗ 
 **Document 1**: Kiss and Tell is a 1945 American comedy film starring then 17-year-old Shirley Temple as Corliss Archer. 
 **Document 2**: As an adult, Shirley Temple was named United States ambassador to Ghana and to Czechoslovakia, and also served as Chief of Protocol of the United States. | Transformer-XH predicts the correct answer document Shirley Temple. However it could not distinguish from the wrong answer ambassador which she was named but not held that position. |

Table 7: Additional examples for model prediction on HotpotQA dataset, the first example is the correct prediction (+), the other two examples are the wrong predictions (-).

research (Ding et al., 2019). We train Transformer-XH and the GNN of CogQA (w. BERT IR) for 2 epochs. All other BERT based models use the default BERT parameters and train the model for 1 epoch.

## A.4   IMPLEMENTATION DETAILS OF COGQA (W. BERT IR)

This section discusses our implementation of CogQA (W. BERT IR). We start with the same documents from BERT IR, the one used by Transformer-XH, and then implement the following steps:

### A.4.1   HOP ENTITY EXTRACTION

For each document from the previous step, we run BERT MRC model and limit the span candidates as hyperlinked entities for hop entity extraction (e.g., in Figure 1, "Harvard University" is a hop entity). Following Ding et al. (2019), we predict the top three entities that above the relative threshold that is the start span probability of [CLS] position.

### A.4.2   ANSWER SPAN EXTRACTION

For each document (add the hop entity document), following Ding et al. (2019), we run BERT MRC model to extract spans (e.g., "Combridge" in Figure 1.). We predict the top one span that above the threshold that is the start span probability of [CLS] position.

We train both hop entity extraction and span extraction tasks with same BERT model but different prediction layers. For each training example, we extract the link between two given supporting pages. The page includes the link (e.g., "Harvard University" in Figure 1.) is the supporting page

for hop entity extraction, while the other page is the answer page (e.g., "Combridge" in Figure 1.) for answer span extraction.

### A.4.3  GAT MODELING

All the entities and answer spans form the final graph. The nodes are the entities and spans, and edges are the connections from the entities to the extracted hop entities or spans.

We use BERT for each node representation with question, anchor sentences and context, following Ding et al. (2019). We run GAT (Veličković et al., 2018) on top of BERT to predict the correct answer span node.

### A.4.4  COMPARISON QUESTIONS

After predicting supporting facts, we concatenate the sentences and follow Min et al. (2019b) to run a BERT MRC model to predict either span or yes/no as the answer.

### A.5  ADDITIONAL CASE STUDY

We provide addition case studies in Table 7. The first case can be directly predicted through the clear evidence chain "the 1925 Birthday Honours"→"George V"→"1865". In the second case, the first document ("Algeria at the FIFA World Cup") has no link to any other documents, therefore the model can not access the correct answer. The third case is more reading comprehension oriented, where the model can not distinguish the correct and wrong spans inside one sentence.

