# OpenReview forum: "Transformer-XH: Multi-Evidence Reasoning with eXtra Hop Attention"
_ICLR.cc/2020/Conference — Accept (Poster)_

### Official Review · AnonReviewer3 · 2019-10-16
**Official Blind Review #3**

**Rating:** 6

**Review:**

This paper introduces the Transformer XH model architecture, a transformer architecture that scales better to long sequences / multi-paragraph text compared to the standard transformer architecture. As I understand, Transformer XH is different from the standard transformer architecture in two main ways: 1) adding attention across paragraphs (or sub-sequences) via the CLS token and 2) defining the structure of that attention based on the entities in the paragraphs (or sub-sequences). The paper tackles an important problem (learning from long sequences) and achieves good empirical results on HotpotQA.

Modification #1 has already been explored by previous works which should be discussed in the paper. "Generating Wikipedia by Summarizing Long Sequences" by Liu, Saleh, et al. propose the T-DMCA model, with a similar motivation: enabling the transformer to work on/scale to longer sequences. T-DMCA and Transformer XH have some difference (T-DMCA seems to have more capacity while Transformer XH is simpler); I think it is necessary to compare against Transformer XH against T-DMCA on HotpotQA to know whether a new architecture is really necessary for HotpotQA. "Generating Long Sequences with Sparse Transformers" from Child et al. also proposes another general transformer architecture that can handle long sequences, and it would be ideal to compare against this architecture as well. Sparse Transformers reduce the time complexity of attention to reduce O(n√ n), which seems similar to the reduction that Transformer XH gets.

For modification #2 (defining the attention structure beforehand using e.g. entity linking), it does not seem too difficult to learn the attention structure directly instead, as confirmed by the ablation in Table 3 which uses a fully connected graph structure. A model that learned the attention pattern or used a fully connected graph would be more general (but more similar to T-DMCA and sparse transformers).

The empirical results are good. It's nice that a simple/straightforward architecture like Transformer XH works quite well (compared to some previous approaches which were not as elegant). However, I do not feel that prior work has explored the best transformer architectures for HotpotQA (such as Sparse Transformers or T-DMCA), and since this work is specifically proposing a new transformer architecture, I think it is important to compare directly against other transformer architectures. Other (previously) SOTA models like SR-MRS are quite simple (conceptually), so it's likely that such models will also be outperformed by transformer architectures that are better adapted to long sequences. In general, I think that the most relevant baseline already present is SR-MRS rather than CogQA; the fact that such a simple approach like SR-MRS works well is an important fact about HotpotQA to take into account (even if SR-MRS is concurrent). In other words, SR-MRS shows that previous models like CogQA are likely weaker baselines.

Because of the missing baselines and limited novelty compared to prior work, I overall lean towards rejecting the paper (despite the good empirical results).

I do have a few specific questions for the authors:
- Would you mind providing further details about the following sentence? "For better retrieval quality, we use a BERT ranker (Nogueira & Cho, 2019) on the set Dir ∪ Del and keep top K ranked ones."
- Is only answer-level supervision used? Or is supporting-fact level supervision used to train any of the rankers or pipeline?

**Experience Assessment:**

I have published one or two papers in this area.

**Review Assessment: Checking Correctness Of Derivations And Theory:**

N/A

**Review Assessment: Checking Correctness Of Experiments:**

I assessed the sensibility of the experiments.

**Review Assessment: Thoroughness In Paper Reading:**

I read the paper thoroughly.

---

> ### Author Response · Authors · 2019-11-09
> **Response to R3**
>
> Thanks a lot for your review!  The revised version has improved writing and below we would like to clarity some confusions.
>
> On the differences from standard Transformer architecture.
> As discussed in Section 1 and 2, the motivation of Transformer-XH is to model multiple documents/pieces of texts not in the sequential format, but is not to handle long text sequences. In Table 3, we have compared with Transformer-XL which we treat the multiple documents as a text sequence. The results are dramatically worse, which is expected because the evidence documents for QA do not have a natural sequential order.
> Defining the structure is not a contribution of this work. As you mentioned, in Table 3, Transformer-XH does not need the pre-given structure and can learn the importance of edges in the fully connected graph. Transformer-XH is the model that learns the attention pattern on a fully connected graph; its performance is almost identical on the fully connected graph or on the Wikipedia graph. We revised Sec 3 to clarify this.
> On the other hand, the pre-given structure we experimented with is not based on the entities in the paragraphs. It is the Wikipedia hyperlink graph that connects the Wikipedia documents. The entity linking is used in the retrieval step.
>
>
>
> On comparison with T-DMCA and Sparse Transformer.
> We compared Transformer-XH with Transformer-XL, which is the state-of-the-art Transformer architecture in modeling long texts. The language model experiment in Section 7.2 of the Sparse Transformer paper shows that Sparse Transformer performs on par with Transformer-XL.  We don’t see a recent comparison of Transformer-XL or Sparse Transformer with T-DMCA on modeling long text.
> It’s worth mentioning that we view Transformer-XH not a universally better but a differently purposed Transformer architecture compared with Transformer-XL and Sparse Transformer. They share the motivation of upgrading Transformer for richer text formats: Transformer-XH focuses on multi-documents where no clear sequential structure exists while the other two focus on long sequential texts. Our experiments in Table 3 show that it is not suitable to treat the retrieved documents as a long sequence of texts.
>
> On whether prior work has explored the best Transformer architectures for HotpotQA.
> Thanks for pointing out this! This argument also applies to other popular NLP datasets such as GLUE. The major reason that prevents exploring the effectiveness of T-DMCA and Sparse Transformer (or other Transformer architectures) is that there is no pre-trained Transformer using these two architectures. The pre-trained Transformers are so effective that we have to somehow build on top of them, to achieve competitive performance on the leaderboard. And to the best of our knowledge, there is no existing pre-trained T-DMCA or Sparse Transformer at the scale of BERT.
> We think this observation points out two key challenges we as a community should address. The first is how to enrich the Transformer architecture on top of pre-trained vanilla Transformer, where Transformer-XH was proposed as one solution to upgrade Transformer with the ability to model multiple documents. The second is what are the best Transformer architectures when used with BERT style pre-training. The latter will require more computing resources and is out of the scope of this paper.
>
> On the proper baseline.
> SR-MRS was posted on ArXiv one week before the paper submission deadline, thus we treat it as contemporary method and briefly compared with it in our submission version and showed the advantage of our Transformer-XH. On our current version, we revise our discussions in Sec. 4 and Sec. 5 to put more comparison with SR-MRS.  SR-MRS focuses on the retrieval stage while Transformer-XH focuses on the reasoning stage after the retrieval, albeit the retrieval methodology we used is close to SR-MRS. We run Transformer-XH on the retrieved documents of SR-MRS and include the results in Table 2: “Transformer-XH (w. SR-MRS)”. Transformer-XH is as effective in the reasoning stage when using SR-MRS's retrieved documents.
>
> On details of the retrieval part:
> Thanks for pointing out this! We include more details of the retrieval part in Appendix A.1 and ablation studies Appendix A.2. Transformer-XH’s effectiveness is robust to different settings and different retrieval systems.
>
> On supporting-fact level supervision:
> We use  supporting-fact level supervision to train the BERT ranker. This is the only place the supervision is used in our system and we follow the previous research in Hotpot QA’s FullWiki setting (CogQA and SR-MRS). We consider this is a realistic setting: In search engines, the ranker is first trained by document level labels and then the QA model is trained on the answer level labels.
>
> We hope our revisions have addressed your questions.  We are happy to discuss if you find any of them remain unclear or have additional questions and will revise our paper accordingly.

---

> > ### Comment · AnonReviewer3 · 2019-11-13
> > **TransformerXH is a way to extend BERT to long context**
> >
> > Thanks for the helpful rebuttal. You make several good points about the differences between Sparse Transformer, T-DMCA, and Transformer XH. In particular, the key novelty of TransformerXH seems to that it is a way to use BERT (or other models pre-trained on shorter contexts) on longer contexts. It would be nice if this would be made clearer in the framing of the paper. This distinction should also be made clear in e.g. the related work (explicitly citing/discussing the connection/difference with SparseTransformer and T-DMCA).
> >
> > I am also excited about the additional results on the FEVER dataset. These results seem to validate that the architecture is quite general (especially compared to many existing HotpotQA models). (It would be nice if these results were moved from the appendix into the main body of the paper.) Because of the FEVER results and the clarification on TransformerXH's motivation, I have increased my score from 3 (weak reject) to 6 (weak accept).
> >
> >
> > Regarding the Transformer XL baseline, I think it is helpful to see as a comparison. However, unless I am mistaken, I think it may not be the best baseline to separate the paragraphs into separate inputs to BERT is sub-optimal; the recurrence connection across more paragraphs will diminish the model's ability to look back at previous paragraphs. I think a better baseline would be to concatenate all paragraphs and use the sliding window approach from the original BERT paper (used on SQuAD). Granted, I think Transformer-XH is likely to scale better (especially as the length of each paragraphs/documents grows, as you can't fit multiple paragraphs/documents into a single window of your transformer model).

---

> > > ### Author Response · Authors · 2019-11-14
> > > **Response to R3 comments**
> > >
> > > Thanks for your update!
> > > - We have added the comparison between our Transformer-XH and other transformer architectures (T-DMCA, Sparse-Transformer etc.) in the related work section. We will change the paper frame in our next version of the paper.
> > > - It is a nice suggestions. We will move the FEVER experiment to the main paper content and perhaps revise our framing that Transformer-XH is general on multiple tasks.
> > > - SR-MRS somehow adopts a close approach that first uses BERT to filter most of the un-relevant paragraphs (and sentences), then concatenate the remaining into the BERT MRC model (it will not exceed the maximum length after the filtering). Their filtering seems to be very effective, albeit we think your suggested method is an interesting baseline. We will try concatenating all the top paragraphs, use the sliding window approach you suggested, and include this baseline in the next version.

---

### Official Review · AnonReviewer2 · 2019-10-22
**Official Blind Review #2**

**Rating:** 8

**Review:**

Summary: This paper introduces a way to train transformer models over document graphs, where each node is a document and edges connect related documents. It is inspired by the transformer-XL model, as well as Graph Neural networks. They apply this model to answer multi-hop questions on the HotPotQA dataset and outperform the previous SOTA. The model particularly improves performance on the bridge style questions of HotPotQA. They are able to do this in a single step, rather than a multi-stage process as done by previous approaches.

Strengths: The model is described in a very detailed manner with contrasts drawn to previous models, which provides excellent motivation for the decisions taken by the authors. I enjoyed reading section 2 as it very succinctly describes previous approaches and introduces transformer-XH. The paper has very detailed and insightful ablation studies, including hop steps and hop attentions, and other graph structures.

Weaknesses:

It took me a lot of effort to figure out that the transformer-XH is only applied to the bridge questions, and part of the overall gains are due to a better retrieval pipeline on the comparison questions. Please explicitly make this clear.

Also, there seems to be a lot of gain even on single-hop questions, and its not clear if overall performance improvement can be attributed to modeling the graph structure, as opposed to other confounding factors. Can the authors elaborate a bit more on why this might improve performance on single hop questions?

Very good evaluation on HotPotQA, but would be even stronger if this were applied to at least one other task/dataset.

Questions:

1. Any intuition as to why the EM performance improvement on single-hop questions is the about the same as the performance improvement on the multi-hop questions (~5%)?

2. In Example 2 in Table 4, it is not clear from the text as to why the BERT pipeline fails to get the correct result. If I understand correctly, both models use the same document graph construction method? Is this not the case? i.e. does the pipeline model have access to the exact same documents that form the Transformer-XH's document graph? Could you explain this cascade error here a bit more?

3. I assume that we can use directed as well as undirected edges in the document graph? Would be good to clarify this.

4. In equations 11 and 12, are you missing normalization operators to specify a distribution, perhaps a softmax?

5. "Our pipeline" in Table 1 is confusing. Would be nice to mention in the caption that it is a baseline constructed using BERT on your retrieval method etc.

**Experience Assessment:**

I have published in this field for several years.

**Review Assessment: Checking Correctness Of Derivations And Theory:**

I carefully checked the derivations and theory.

**Review Assessment: Checking Correctness Of Experiments:**

I carefully checked the experiments.

**Review Assessment: Thoroughness In Paper Reading:**

I read the paper thoroughly.

---

> ### Author Response · Authors · 2019-11-09
> **Response to R2**
>
> Thanks a lot for your review!
>
> On the clarity of the gains on bridge questions:
> Thanks for your comments! It motivates us to evaluate our Transformer-XH on comparison questions. On Table 2, Transformer-XH consistently improves over SR-MRS and  CogQA (W. BERT IR) on comparison questions using the same retrieved documents. On the bridge questions, we compare SR-MRS and Transformer-XH using the same retrieved documents on Table 2, the big improvements illustrate the stronger modeling capacity of Transformer.
>
> Q1. On the improvement on single-hop questions:
> All Hotpot QA questions, by design, are multi-hop or multi-evidence questions. The questions are written by mTurkers to ask for information from two provided Wikipedia documents. As a result, the two documents should contain useful information to answer the question. The single-hop / multi-hop were categorized by [1]. They consider a question is single-hop if the single-hop BERT MRC has non-zero F1 when used on individual documents. This is a rough estimation on the distractor setting, as in the FullWiki setting the other documents might still provide useful information, which is better leveraged by Transformer-XH to provide more accurate answers. We have revised the discussion of Table 2 to reflect that the single-hop / multi-hop is a rough estimation.
> On the other hand, as shown in our new FEVER results, Transformer-XH performs much better on multi-evidence cases, with 20 absolute point gains. On FEVER the claims are grouped by human labels and the single-evidence claims do only need one evidence sentence to verify. On those single evidence cases, Transformer-XH performs slightly better than recent approaches.
>
>
>
>
> Q2 on the BERT pipeline cascade error.
> Both the pipelined approach and Transformer-XH starts with the same set of documents (D_ir and D_el in Section 3). Then the pipelined approach starts with extracting hop entities using BERT MRC model for each document, (e.g., Harvard from Facebook’s Wikipedia documents), then it expands the document set by adding the documents corresponding to the hop entities and extracts the answer span for the new document set.
> In the second example of Table 4, the cascade error is from the hop entity extraction, without the information from the other documents, the MRC model can not distinguish which is the correct hop entity among the names [Frank Lowy], [Ben Buckley], [Quentin Bryce] and [Elle Macpherson].
> Another interesting finding is that the training data only teaches the MRC model to find the best span in each document, not the top K possible ones. We use the threshold based approach (if we choose Top-K, the constructed graph is too big to fit into the memory), and on this example, the other entities are below the threshold and are filtered.
>
> Q3 on graph structure.
> Yes, we can use undirected graph as well. We have updated the description of evidence graph edges in Section 3 to reflect this.  In the ablation study of Table 3, bottom left, we show that Transformer-XH performs similar with undirected edges or directed edges, fully connected graph or Wikipedia graph. The studies of Figure 2 on the Fully Connected Graph show Transformer-XH distinguishes the importance of edges connecting different documents.
> Q4 on equation 11 and 12.
> Thanks for spotting this! We use softmax after all linear layers, similar to other MRC methods. We fixed this in the revised version.
> Q5 on method name.
> Thanks for your suggestion! We rename it to “CogQA (w. BERT IR)” to reflect that it is our implementation of CogQA but with evidence documents retrieved by our BERT based ranking component. We update Section 4 to make this clear.
>
> On performance on another task.
> We add an experiment on the FEVER dataset, following a recent paper on the top of the leaderboard that posted on ArXiv (https://arxiv.org/pdf/1910.09796.pdf).  We use their released retrieved sentences (Top 5 from BERT), build a fully-connected graph, where each node is retrieved sentence, and apply Transformer-XH on the graph using the same parameter settings used on Hotpot QA.  Transformer-XH achieves similar results as this contemporary work, showing its robust effectiveness. We include the details of FEVER evaluation in Appendix A.6.
>
> [1] Min, Sewon, et al. "Multi-hop Reading Comprehension through Question Decomposition and Rescoring." arXiv preprint arXiv:1906.02916 (2019).

---

> > ### Comment · AnonReviewer2 · 2019-11-15
> > **Thanks for the detailed responses and results on a new dataset.**
> >
> >
> > I'm still a little confused about the comparison between Transformer-XH and CogQA.
> >
> > If I understand correctly from your description of CogQA and your responses, CogQA uses Bert-MRC to create its own version of D_exp, which is different from the D_exp that you construct for Transformer-XH.
> >
> > Is there any way to compare the methods using the same set of documents including D_exp?
> >
> > Put another way, it seems that the 2nd level documents i.e. the documents linked to the documents retrieved in the first pass, are different for transfomer-XH and Cog-QA (Please correct me if I'm wrong).  If this is the case, its hard to separate the gains solely due to X-fmer-XH from the gains from the evidence graph contruction step.
> >
> > Put yet another way, is there some hindrance that prevents you from using the same documents that were used to create your evidence graph (including D_exp), in the CogQA model?
> >
> > Thanks!

---

> > > ### Author Response · Authors · 2019-11-15
> > > **Response to R2 comments**
> > >
> > > Great point! We actually have tried this idea when developing Cog QA (w BERT IR). We use D_exp instead of hop entity extraction results, then run the BERT MRC to extract candidate spans and use GNN to rank the spans, following CogQA. The preliminary results are worse than Cog QA (w BERT IR).
> > > Recall that the GNN in CogQA focuses on ranking the candidate spans where the span extraction component (second hop MRC) does not have access to the global information of all the evidence documents. If we don’t filter D_exp using the hop entity extraction (first hop MRC) , the second hop MRC provides more noisy spans and is further to confuse the GNN. As a result, this leads to worse performance in both effectiveness and efficiency. Although the first hop BERT introduces more cascade errors, it is necessary for the latter part of the CogQA pipeline to be effective.
> > > We guess this is the reason the CogQA authors use the first hop BERT MRC to filter down entities early in the pipeline. It is a reasonable design choice as they used multiple single-hop BERT MRC to mimic the multi-hop process.
> > > On the other hand, Transformer-XH jointly learns to represent and model the entire evidence document sets; thus it avoids such cascade errors and is a simpler and more effective model.

---

### Official Review · AnonReviewer1 · 2019-10-24
**Official Blind Review #1**

**Rating:** 6

**Review:**

The paper is proposing an extension of the Transformer matching and diffusion mechanism to multi-document settings.
To do so, the authors introduce a special representation for gathering document level information which is then used for propagation among documents latent representations.
The extension seems quite simple and natural.
The method is evaluated on multi-hop machine reading over the hotpotqa dataset in the Fullwiki settings.
However, it could have made sense to evaluate the method in the distractor settings too.
In this context, the evidence graph where the model is trained is built using the canonical retrieval technique.
Then, the method is using a pre-trained NER model to extract entities on the question and the candidate documents on Wikipedia for matching.
Finally, a BERT ranker model is used to re-rank the retrieved candidate documents.
The proposed method seems to heavily dependant on this hand-crafted extraction process.
Unfortunately, one concern is that the reasoning model, while been quite original, is not tested in large scale retrieval cases to assess its robustness.
Indeed, the number of retrieved documents to create the evidence graph seems to not have been mentioned.
The method improves the current state of the art.

**Experience Assessment:**

I have published in this field for several years.

**Review Assessment: Checking Correctness Of Derivations And Theory:**

I carefully checked the derivations and theory.

**Review Assessment: Checking Correctness Of Experiments:**

I carefully checked the experiments.

**Review Assessment: Thoroughness In Paper Reading:**

I read the paper thoroughly.

---

> ### Author Response · Authors · 2019-11-09
> **Response to R1**
>
> Thanks a lot for your review!
>
> On hand-crafted retrieval:
> The focus of this paper is on learning the representation of texts with non-sequential structures. We keep our retrieval method similar to previous solutions and didn’t claim novelty from it. We revised the Evidence Graph Construction paragraph in Section 3 and Appendix A.1 to include more details of the retrieval step. We also added ablation studies on the different number of documents used in the evidence graph in Appendix A.2.
> As shown in Table 2, Transformer-XH's effectiveness is due to the novel design of information flow between documents, not how the documents are retrieved: it works on par/even better when directly applied to the top documents retrieved by SR-MRS, which were recently shared by their authors. On Table 5, we show Transformer-XH’s effectiveness is stable with different numbers of evidence documents on Hotpot QA bridge questions.
> We agree that this is a key challenge in open domain QA. Retrieving the candidate documents is the core ad hoc retrieval task which now, unfortunately, still has to use the cascade pipeline. For example, a recent state of the art method in the MSMARCO ranking task uses a similar multi-stage retrieval and BERT reranking: https://arxiv.org/abs/1910.14424.
>
> On FullWiki VS Distractor Setting:
> We had concerns in the distractor setting as previous research [1] found that the distractor documents are too weak and reduce the need of multi-hop reasoning. Our empirical studies find that, because the negatives are from an unsupervised retrieval model, it is fairly easy for the BERT ranker to pick the two ground truth paragraph from the negative distractors. In addition, the construction of Hotpot QA often mandates the two ground truth paragraphs are connected by a Wikipedia link. This may leak some ground truth information as the distractors are often not connected. The Fullwiki setting is more robust because 1) the negative paragraphs are much stronger distractors from the BERT ranking model; 2) there are many negative paragraphs connected by Wikipedia links when constructing the evidence graph. We focus on the FullWiki setting which is more challenging and realistic.
> We provide the evaluation of Transformer-XH on the FEVER fact verification dataset in Appendix A.7, the other task SR-MRS was applied.  The results further demonstrate the robust effectiveness of Transformer-XH.
>
> On large scale retrieval:
> The corpus we retrieve documents from is the full Wikipedia corpus. Though it is not as large scale as CommonCraw or ClueWeb, the full Wikipedia corpus (more than 5 million) is not small and is a common corpus used in many other NLP datasets, e.g., SQuAD, Natural Questions, FEVER etc..
> Moving towards more realistic open domain settings where the documents are general web documents is a next step on scale up the impact of multi-hop QA. How to construct such multi-hop large scale open domain QA dataset is an important future research direction itself. Nevertheless, we think being able to answer from Wikipedia still covers a lot of questions from search engine, e.g., Google’s Natural Questions dataset is built based on real queries and also uses Wikipedia as evidence source.
>
>  [1] Min, Sewon, et al. "Compositional Questions Do Not Necessitate Multi-hop Reasoning." arXiv preprint arXiv:1906.02900 (2019).

---

### Author Response · Authors · 2019-11-09
**Updates in the Paper Revision**

We would like to thank reviewers for their reviews!  We have uploaded a revised version of the paper.

The updates include:
1) new results using SR-MRS's retrieved documents in Table 2, showing Transformer-XH's robust effectiveness with different IR systems;

2) Experiments on fact verification (FEVER dataset ) in Appendix A.6, which further demonstrate Transformer-XH effectiveness, especially on claims that require multiple evidence sentences;

3) Revised Sec 3 with more details in evidence construction and discusses that Transformer-XH can use fully-connected graph and learn the edge weights automatically;

4) Renaming BERT Pipeline to CogQA (w. BERT IR) which better reflects what the method does;

5) Revised Appendix A.1 with more details on the BERT IR system;

6) Revised Appendix A.2 with new ablation studies on the influence of the retrieval stage.

7) Revised related work with comparison to other Transformer architectures.

---

### Decision · Program_Chairs · 2019-12-19

**Decision:**

Accept (Poster)

**Comment:**

This work examines a problem that is of considerable interest to the community and does a good job of presenting the work. The AC recommends acceptance.